# A Rapid, Sensitive, Low-Cost Assay for Detecting Hydrogenotrophic Methanogens in Anaerobic Digesters Using Loop-Mediated Isothermal Amplification

**DOI:** 10.3390/microorganisms8050740

**Published:** 2020-05-15

**Authors:** Anna M. Alessi, Bing Tao, Wei Zhang, Yue Zhang, Sonia Heaven, Charles J. Banks, James P. J. Chong

**Affiliations:** 1Department of Biology, University of York, Wentworth Way, York YO10 5DD, UK; anna.alessi@york.ac.uk; 2Biorenewables Development Centre Ltd., 1 Hassacarr Close, Chessingham Park, Dunnington, York YO19 5SN, UK; 3Water and Environmental Engineering Group, Faculty of Engineering and Physical Sciences, University of Southampton, University Road, Southampton SO17 1BJ, UK; Bing.Tao@soton.ac.uk (B.T.); Wei.Zhang@soton.ac.uk (W.Z.); y.zhang@soton.ac.uk (Y.Z.); S.Heaven@soton.ac.uk (S.H.); C.J.Banks@soton.ac.uk (C.J.B.)

**Keywords:** hydrogenotrophic methanogens, anaerobic digestion, loop-mediated isothermal amplification, metagenomes

## Abstract

Understanding how the presence, absence, and abundance of different microbial genera supply specific metabolic functions for anaerobic digestion (AD) and how these impact on gas production is critical for a long-term understanding and optimization of the AD process. The strictly anaerobic methanogenic archaea are essential for methane production within AD microbial communities. Methanogens are a phylogenetically diverse group that can be classified into three metabolically distinct lineages based on the substrates they use to produce methane. While process optimization based on physicochemical parameters is well established in AD, measurements that could allow manipulation of the underlying microbial community are seldom used as they tend to be non-specific, expensive, or time-consuming, or a combination of all three. Loop-mediated isothermal amplification (LAMP) assays combine a simple, rapid, low-cost detection technique with high sensitivity and specificity. Here, we describe the optimization of LAMP assays for the detection of four different genera of hydrogenotrophic methanogens: *Methanoculleus, Methanothermobacter, Methanococcus,* and *Methanobrevibacter* spp. By targeting archaeal elongation factor 2 (aEF2), these LAMP assays provide a rapid, low-cost, presence/absence indication of hydrogenotrophic methanogens that could be used as a real-time measure of process conditions. The assays were shown to be sensitive to 1 pg of DNA from most tested methanogen species, providing a route to a quantitative measure through simple serial dilution of samples. The LAMP assays described here offer a simple, fast, and affordable method for the specific detection of four different genera of hydrogenotrophic methanogens. Our results indicate that this approach could be developed into a quantitative measure that could provide rapid, low-cost insight into the functioning and optimization of AD and related systems.

## 1. Introduction

In addition to its vital and well-established role in stabilization of wastewater biosolids, anaerobic digestion (AD) is now increasingly used for resource recovery from a wide variety of agri- and food-processing wastes and industrial effluents. Key to this is the anaerobic microbial community that facilitates the degradation of organic materials through hydrolysis of large biopolymers. Hydrolysis of polymers leads, via a cascade of metabolic pathways, to the production of small molecules and ultimately to carbon dioxide and methane, the main energy-carrying component of biogas, and a nutrient-rich digestate residue [1]. This process is mediated by hundreds of different interacting microbial species. The presence, absence, and abundance of different genera affects the specific metabolic functions or services available to the overall microbial community. Understanding how these functions impact on the interlinked processes of solids degradation and gas production offers new opportunities for optimization and enhancement of conventional AD. AD processes will also be an essential element in future biorefineries, and greater understanding of these community synergies and interdependencies may lead to new technologies based on exploiting the metabolic versatility of anaerobic consortia for generation of bio-based building block products.

One critical component of the microbial community in AD for methane production is the strictly anaerobic methanogenic archaea. Methanogens are a phylogenetically diverse group that can be classified into three metabolically distinct lineages (hydrogenotrophic, acetoclastic, and methylotrophic, [2]) based on the substrates used to produce methane. Hydrogenotrophic methanogens (Equation (1)) reduce CO_2_ to CH_4_ using H_2_. The acetoclastic methanogenesis pathway is responsible for two-thirds of biogenic methane production, but it has been reported only for organisms within the order *Methanosarcinales* (Equation (2)), while methylotrophs (Equation (3)) can generate CH_4_ from a variety of chemical compounds containing a methyl group such as methanol, methylsulfides, or methylamines.
4 H_2_ + CO_2_ → CH_4_ + 2 H_2_O, ΔGo′ = −131 kJ mol^−1^(1)
CH_3_COOH → CH_4_ + CO_2_, ΔGo′ = −36 kJ mol^−1^(2)
CH_3_OH + H_2_ → CH_4_ + H_2_O, ΔGo′ = −112.5 kJ mole^−1^(3)

In typical wastewater biosolids digesters, the acetoclastic and hydrogenotrophic pathways dominate methane production and are usually associated with particular genera. For example, all members of the order *Methanosarcinales* (including the genera *Methanosarcina*, *Methanosaeta,* and *Methanolobus*) contain genes for *c*-type cytochrome and methanophenazine (a functional menaquinone analogue) that enable them to utilize a broad range of substrates including hydrogen, formate, and acetate. In contrast, hydrogenotrophic methanogens of the orders *Methanopyrales*, *Methanococcales* (e.g., genus *Methanococcus*), *Methanobacteriales* (e.g., genera *Methanothermobacter*, *Methanobrevibacter*), and *Methanomicrobiales* (e.g., genus *Methanoculleus*) do not contain a cytochrome system and are limited to growing mainly on H_2_ and CO_2_, although many can also grow on formate [2,3]. Hydrogen-scavenging methanogens can partner with syntrophic acetate-oxidizing bacteria (SAO) to support stable methane production in AD under certain conditions [4]. The presence and relative abundance of hydrogenotrophs is of particular importance for the reductive biomethanisation of CO_2_ from both inorganic and organic sources, e.g., as a means of in situ biogas upgrading [5] or for ex situ carbon capture and utilization [6]. Recent studies reported microbial community shifts from *Methanoculleus* and *Methanoregulaceae* species towards hydrogenotrophic methanogenesis when ex situ configurations were employed under thermophilic and mesophilic conditions, respectively [7]. Similarly, Kougias et al. [8] observed hydrogen-assisted methanogenesis with *Methanothermobacter thermautotrophicus* as the dominant member of the archaeal community in ex situ reactors. Other studies showed that hydrogenotrophs are more tolerant to high ammonia concentrations than acetoclastic methanogens and hence more abundant in digesters treating high-ammonia feedstocks [9].

While physicochemical monitoring is well established as a basis for process optimisation in AD, measurements indicating the presence/absence of particular microbial community members, or that quantify these and could facilitate manipulation of the underlying microbial community, tend to be non-specific, expensive, and time-consuming, or a combination of all three. Culturing methods for anaerobic species are relatively inexpensive but require sophisticated infrastructure and are time consuming as they rely on the growth rates of the organisms being assessed. This is compounded by our inability to grow many species in isolation [10]. Molecular methods such as PCR-based techniques (including qPCR) are sensitive but require expensive equipment and reagents and take a number of hours. Approaches such as 16S rRNA amplicon or whole metagenome high-throughput sequencing provide excellent specificity, but are also expensive, slow (days to weeks), and require a high level of expertise for data analysis and interpretation.

Loop-mediated isothermal amplification (LAMP) assays combine a simple, rapid, low-cost detection technique with high sensitivity and specificity. LAMP assays are an efficient diagnostic tool, can be easily performed as they require a minimal amount of equipment, and utilize low-cost reagents. LAMP assays are widely employed in the food industry and in clinical settings where they have been used to detect specific pathogens including bacteria, parasites, fungi, and viruses [11,12]. LAMP relies on identifying a conserved gene within a targeted group of microorganisms against which a series of four to six primers must be designed (Figure 1). The assay is performed, often in less than one hour, by amplifying the target gene using Bst polymerase under isothermal conditions. A range of detection methodologies have been reported including visual identification of turbidity, colorimetric or fluorescent DNA interchelators, and end-point agarose gel electrophoresis [12].

This work aimed to determine the feasibility of using LAMP assays for rapid generation of diagnostic data on the status of a methanogenic microbial community with the notion that this information could be used to inform and improve decisions regarding process control in AD and related processes. We identified a candidate target gene to test the ability of LAMP to provide discriminating information on hydrogenotrophic methanogens. The suitability of this gene for our application was evaluated in silico and confirmed in vitro by PCR amplification of the external primer pair before being deployed in LAMP assays. We compared the results we gained from other methanogen-containing samples such as freshwater sediment, human stool samples, and soil to samples from AD systems to further demonstrate the discriminatory power of our assays.

## 2. Materials and Methods

### 2.1. Identification of LAMP Assay Biomarkers

An unpublished AD metagenomics dataset (BTR1 A-H and BTR2 A-H, part of PRJEB27206 project submitted to the European Nucleotide Archive with sample accession numbers ERR2642234–ERR2642249) sequenced using the Illumina HiSeq platform was used to identify candidate target genes for the LAMP assay. Briefly, reads were assembled with MEGAHIT [13], ORFs (*n* = 230,339) were predicted using PROKKA [14] and assigned KEGG ontology and taxonomy using GhostKoala [15]. KEGG orthologues identified uniquely as archaeal (*n* = 434) were further examined by UniProt [16] to identify proteins belonging to different hydrogenotrophic methanogen orders (*Methanomicrobiales*, *Methanobacteriales,* and *Methanococcales*). These were aligned using UniProt and identity between proteins was noted (April 2020). Candidate gene nucleotide sequences from different methanogenic genera (*Methanoculleus* sp., *Methanothermobacter* sp., *Methanococcus* sp., and *Methanobrevibacter* sp.) were retrieved from the NCBI database. Predicted ORF homologues from our unpublished assembly were also included in the analysis. Nucleotide sequences were aligned using CLC Genomics Workbench (v.8.5.1, QIAGEN, Redwood City, CA, USA). Highly conserved regions of genes were identified by visual inspection of the alignment and selected based on a threshold of >80% identity per nucleotide position to establish nucleotide consensus sequences (Appendix A).

### 2.2. LAMP Primer Design

Consensus sequences of conserved regions of target genes were uploaded to Primer Explorer V5 [17] and LAMP primers were designed using the default settings (length: F1c/B1c = 20–22 bp, F2/B2 = 18–20 bp, F3/B3 = 18–20 bp, Tm: F1c/B1c = 64–66, F2/B2 = 59–61, F3/B3 = 59–61, GC rate (%) = 40–65, dG threshold: 5′ stability = −3, 3′ stability = −4, dimer check—−2.5, distances: F2-B2 = 120–180, Loop(F1c-F2) 40–60, F2-F3 = 0–20, F1c-B1c = 0–100, Table 1). Loop primers were designed by Primer Explorer V5 or manually by visual evaluation of the archaeal elongation factor 2 (aEF-2) genes alignments (Appendix A). Primer sequences were searched against the NCBI database to confirm their specificity for selected hydrogenotrophic genera. Desalted primers were synthesised by Integrated DNA Technologies, BVBA (Leuven, Belgium). Primers were re-suspended in sterile nuclease-free water upon arrival and stored in 50 μL aliquots at −20 °C. 

### 2.3. Template DNA

Bacterial cultures of *Escherichia coli* ATCC11775, *Staphylococcus epidermidis* ATCC14990, and *Bacillus* sp. 3PL were grown in 5 mL nutrient broth cultures overnight at 37 °C with shaking. Genomic DNA was extracted using the DNeasy UltraClean Microbial kit (QIAGEN #12224). *Methanoculleus marisnigri* JG1 were grown under anaerobic condition in the medium 141 as described previously [18]. *Methanococcus maripaludis* S1 was cultured in McCas liquid media as described previously [19]. *Methanothermobacter thermautotrophicus* (DSMZ 1053) cultures were grown under chemoautotrophic conditions [20], with H_2_ and CO_2_ as the sole energy and carbon sources. Genomic DNA from *Methanoculleus marisnigri* JG1, *Methanothermobacter thermautotrophicus,* and *Methanococcus maripaludis* S1 was extracted from 10 mL cultures using a DNeasy UltraClean Microbial kit (QIAGEN #12224) with initial bead beating increased to 20 min. Environmental DNA (eDNA) was extracted from 200 mg of biomass using a DNeasy PowerSoil kit (QIAGEN #12888) from samples collected from a 1686 m^3^ anaerobic digester operated by a local water company (NAB), a lab-scale (1 L) mesophilic anaerobic digester with hydrogen addition for in situ CO_2_ biomethanisation (BTR, University of Southampton), a lab-scale thermophilic anaerobic digester fed on food waste (EX, University of Southampton), garden topsoil, and salt marsh sediment samples (F30 and S30). Stool samples (P) were collected from human volunteers with approval from the Biology Ethics Committee (University of York). Metagenomic DNA was extracted from 250 mg stool using the QIAGEN QIAamp PowerFecal DNA kit (#12830). The concentration of genomic and metagenomic DNA was measured using a Nanodrop-3000. Template samples were diluted to 10 ng/μL with sterile, nuclease-free water. Samples were stored at −20 °C.

### 2.4. PCR with Outer F3/B3 Primers

First, 25 μL reactions consisting of 5 μL 5× reaction buffer, 0.5 μL 10 mM dNTP, 1.25 μL 10 μM each forward and reverse primers, 0.25 μL Phusion polymerase (2000 U/mL, #M0530S, New England Biolabs UK, Hitchin, UK), 2 μL template (10 ng/μL), and sterile nuclease-free water (NEB) were assembled on ice. Initial denaturation was performed at 98 °C for 30 s, followed by 25 cycles of denaturation at 98 °C for 10 s, annealing at various temperatures for 10 s, and extension at 72 °C for 15 s. A final extension was performed at 72 °C for 5 min. Products were examined by 1.5% (*w/v)* agarose gel electrophoresis in 1× TAE separated at 110 V for 45 min. PCR amplification products were purified (DNA Clean and Concentrator-25 kit, #D4005, Zymo Research, Irvine, CA, USA) and sequenced using the F3 (10 μM) outer primer using the GATC Supreme Run service (Eurofins Genomics UK, Wolverhampton, UK). Sequencing products were analysed using BLASTn [21] (Appendix A).

### 2.5. LAMP Assay

LAMP reactions were prepared in single 0.2 mL PCR tubes in a laminar-flow hood. Each experiment was repeated independently three times. A 10× primer mix of LAMP primers consisting of 2 μM F3 and B3 primers, 16 μM FIP and BIP primers, and 4 μM FL and BL primers was prepared, aliquoted, and stored at −20 °C. Bst 3.0 reactions consisted of 2.5 μL 10× reaction buffer, 1.5 μL 100 mM MgSO_4_, 3.5 μL 10 mM dNTPs, 2.5 μL 10× primer mix, 1 μL Bst 3.0 enzyme (NEB, #M0374S), and 2 μL 10 ng/µL DNA. Bst 2.0 Warm Start reactions consisted of 12.5 μL 2× Warm Start master mix (NEB #E1700S), 2.5 μL 10× primer mix, and 2 μL 10 ng/µL DNA. All reactions were assembled on ice. A no-template control was included to ensure amplification specificity. Reactions were incubated at either 65 °C for 30 min (Mth, Mco), 65 °C for 45 min (Mcu), or 58 °C for 45 min (Mbr) before the enzyme was heat inactivated at 80 °C for 5 min. Product detection was performed by carefully transferring 9 μL of the stopped reaction to a fresh 0.2 mL PCR tube and adding 1 μL of 1000× SYBR Green I dye (Thermo Fisher Scientific, Waltham, MA, USA). Visual inspection was performed with positive reactions turning yellow/green while negative reactions remained orange. DNA template amplification was additionally visualised by loading 5 μL of product on a 1.5% (*w/v*) agarose gel in 1× TAE following electrophoresis at 110 V for 45 min.

### 2.6. LAMP Sensitivity

First, 10 ng/µL of genomic DNA was diluted by 10-fold serial dilution using sterile nuclease-free water to a concentration of 0.001 ng/µL. LAMP reactions with Mcu and Mco primers were performed in technical duplicates using 2.5 μL 10× reaction buffer, 1.5 μL 100 mM MgSO_4_, 3.5 μL 10 mM dNTPs (source), 2.5 μL 10× primer mix (see LAMP assay section), 1 μL Bst 3.0 enzyme (NEB, #M0374S), and 1 μL (10 ng/μL–0.001 ng/μL) DNA. Bst 2.0 Warm Start reactions for Mth consisted of 12.5 μL 2× Warm Start master mix (NEB #E1700S), 2.5 μL 10× primer mix, and 2 μL 10 ng/µL DNA. Reactions were incubated at either 65 °C for 30 min (Mth, Mco) or 65 °C for 45 min (Mcu) before the enzyme was heat inactivated at 80 °C for 5 min. End-point products were visualised as described in the LAMP assay section.

## 3. Results

### 3.1. aEF-2 Is a Potential Biomarker for Hydrogenotrophic Methanogen Detection in LAMP Assays

To identify hydrogenotrophic methanogen sequences, we sampled and sequenced a lab scale anaerobic digester receiving a supplementary feed of hydrogen to promote in situ CO_2_ biomethanisation [5]. The resulting metagenomic dataset of 225,144,076 reads was assembled using the pipeline described above into 136,759 contigs from which we identified 230,339 putative open reading frames. We identified 434 KEGG orthologues uniquely present in archaea from the ORFs. These orthologues included assignments to cellular processes (*n* = 6), environmental information processing (*n* = 24), genetic information processing (*n* = 139), and metabolism (*n* = 132).

Archaeal elongation factor 2 (aEF-2, aka *fusA*, K03234) was identified as a good target gene for the LAMP assay based on a number of highly conserved regions across the gene that facilitated primer design. In particular, a high level of protein sequence identity (>40%) was observed within different methanogen orders. A Uniprot alignment of the 24 available *Methanobacteriales* aEF-2 protein sequences showed 47% identity, which included 356 identical positions. Similarly, an alignment of the 17 available aEF-2 *Methanococcales* sequences showed 61% identity with 447 identical positions, and *Methanomicrobiales* (*n* = 9) aEF-2 proteins shared 61% identity with 449 identical positions. We included novel putative aEF-2 sequences in our analyses that we identified from our metagenomic assembly as related to the genus *Methanoculleus* (Appendix A, designated as CODx_xxxx). We confirmed that the aEF-2 gene was a good LAMP assay candidate by comparing these results to similar searches that we performed for other potential gene targets including the following genes involved in methanogenesis pathways: methyl-CoA reductase (*mcrA*), methylene tetrahydromethanopterin reductase (*mer*), energy-converting hydrogenase A (*ehaA*), the 30S ribosomal protein S28e (*rps28e*), and 50S ribosomal protein L44e (*rpl44e*) (Table 2). 

### 3.2. Genus-Specific PCR Products Are Generated with LAMP Outer Primers

To demonstrate that the outer LAMP primers were unique and specific to the aEF-2 genes in targeted methanogenic species, the outer forward (F3) and reverse (B3) primers were used in a standard PCR reaction. Three pairs of primers that were designated as the outer primers for future LAMP assays designed to target selected *Methanoculleus, Methanothermobacter,* and *Methanococcus* species (primers F3/B3_Mcu, F3/B3_Mth, and F3/B3_Mco) were shown to correctly amplify aEF-2 genes when provided with gDNA extracted from single species cultures (Figure 2a–c). Sequencing of these products confirmed that only aEF-2 genes were amplified (Appendix A). No amplification of bacterial negative controls (*Escherichia coli* ATCC11775, *Staphylococcus epidermidis* ATCC14990, and *Bacillus* sp. 3PL) was observed with primers that targeted *Methanoculleus* and *Methanococcus* species (Figure 2a,c). Primers targeting *Methanothermobacter* species aEF-2 showed non-specific amplification with template DNA from *Escherichia coli* and *Methanoculleus marisnigri* (Figure 2b, lane 5). Several weak bands were also noted in the eukaryotic negative control performed with F3/B3_aEF2_Mth primers (Figure 2b, lane 7). In the absence of a pure culture of *Methanobrevibacter*, eDNA extracted from eight human stool samples was used as a template as this organism is expected in about one in three of the population. Amplification was observed for two samples indicating that the F3/B3 Mbr primers target an aEF-2 gene (Figure 2d). The products generated by pair F3/B3 primers, were sequenced and confirmed correct affiliation with selected methanogenic species (Appendix A). 

### 3.3. Specificity of Full LAMP Assays

The same template DNAs used to test the outer primers in PCR were used to further examine the specificity of our designed primers in full LAMP assays. LAMP primer sets correctly amplified aEF-2 genes from specific methanogen genomic DNA, producing a characteristic ladder-like pattern. Bacterial and eukaryotic DNA control reactions showed no amplification, indicating that the primers were specific for selected species of methanogens (Figure 3). 

### 3.4. A Set of LAMP Assays to Discriminate between Hydrogenotrophic Methanogenic Genera

We searched our genus-targeted LAMP primer sets against the NCBI database using BLAST adjusted for short sequences. In a few cases, specifically where sequences made use of a high level of degenerate nucleotides, we found that individual primers could potentially anneal to aEF-2 genes from more than one genus. We also noted that some primers could potentially anneal to elongation factor 2 genes from various eukaryotic species. These were discounted as insignificant, as our earlier results (Figure 2 and Figure 3) demonstrated no amplification of eukaryotic EF-2 genes with either outer primer pairs or whole primer sets. Overall, our *in silico* evaluation showed that the majority of our primers theoretically annealed to aEF-2 genes only from the targeted genera (Figure 4).

Complete LAMP reactions were used to screen a range of samples for the presence of hydrogenotrophic methanogenic genera using two different Bst formulations. We extracted DNA from samples of mesophilic and thermophilic anaerobic digesters and compared these to DNA extracted from salt marsh sediment, soil, and human stool samples. A characteristic ladder-like pattern using Bst 3.0 indicated that *Methanoculleus* species were detected in mesophilic anaerobic digester samples (Figure 5a). Assays for detection of *Methanothermobacter* species were initially performed using Bst 3.0. This led to false-positive amplification in non-template controls. Reagent contamination resulting in false positive signals has previously been identified as a potentially confounding factor in LAMP assays [22]. Even after replacing all reagents, the Bst 3.0 chemistry continued to provide inconclusive results when combined with the *Methanothermobacter* primers. In contrast, the Bst 2.0 Warm Start DNA polymerase showed a distinct ladder-like pattern for *Methanothermobacter* positive controls and reliably detected *Methanothermobacter* species in thermophilic anaerobic digester samples (Figure 5b). *Methanococcus* was not detected in any of the environmental samples we tested but was successfully amplified using gDNA isolated from *Methanococcus maripaludis* used as a positive control with both Bst 3.0 and Bst 2.0 Warm Start polymerases (Figure 5c). Reaction conditions for *Methanobrevibacter* required further optimization. As we did not have pure cultures available as a source of genomic DNA for this genus, environmental samples that showed positive amplification with the outer F3/B3 primers (Figure 2 and Figure 3d) were used to optimise the LAMP assays for these species. A 55–65 °C temperature gradient yielded an optimal assay temperature of 58 °C for *Methanobrevibacter* LAMP using Bst 3.0 polymerase due to a lower melting temperature for the F3 and B3 primers. A positive ladder-like pattern was observed using metagenomic DNA extracted from human stool samples (Figure 5d).

### 3.5. LAMP Sensitivity

Genomic DNA from the methanogens *Methanoculleus marisnigri* JG1, *Methanothermobacter thermautotrophicus,* and *Methanococcus maripaludis* was used to determine the potential sensitivity of our methanogen LAMP assays (Figure 6). Decreasing amounts of gDNA from *M. marisnigri*, *M. thermautotrophicus,* and *M. maripaludis* cultures were used to evaluate the sensitivity of individual LAMP assays for detecting these hydrogenotrophic methanogen genera. *Methanoculleus* and *Methanococcus* detection was successful for samples ranging from 10 ng/uL (10 ng) of gDNA to 0.001 ng/uL (1 pg). The *Methanothermobacter* detection limit was 0.1 ng/uL. *Methanobrevibacter* primer sensitivity was not determined as gDNA for this genus was not available. 

## 4. Discussion

In this study, we aimed to develop a low cost, simple, qualitative approach to the detection of selected hydrogenotrophic methanogens in samples from anaerobic digesters operating under a variety of conditions. These protocols could be equally applied to other samples likely to contain methanogens such as environmental microbiomes or high-rate reactors for ex situ CO_2_ biomethanisation. Our LAMP assays provide a rapid indication of the presence or absence of targeted methanogenic genera and do not rely on expensive equipment or extensive post-assay analysis as might be the case with more traditional sequence-based methods. We targeted genera with multiple cultured representatives and publicly available genome sequences to aid primer design. Target genes for LAMP assays require a high degree of conservation at the nucleotide level to facilitate primer design and should ideally be single copy genes that are unique to the microbial group of interest. Gao and Gupta [23] reported 31 proteins uniquely found in various methanogenic archaea. These proteins include well-characterized biomarkers such as enzymes encoded by the *mcr* (methyl coenzyme M reductase [24]) and *mtr* (methyl-H_4_MPT coenzyme M methyltransferase [25]) genes involved in methanogenesis and several hypothetical proteins with domains of unknown function. A lack of sequence data in public repositories for the hypothetical genes precluded their use in LAMP assay design. By conducting comparative alignments of different proteins (Table 2) from selected genera of hydrogenotrophic methanogens, we identified archaeal elongation factor 2 (aEF-2) as a good candidate for LAMP. EF-2 has been used in several studies to evaluate the relationships between archaea, bacteria, and eukaryotes [26,27,28,29]. More recent studies showed EF-2 paralogs in various archaea and eukaryotes suggesting that EF-2 might not be an effective single copy marker gene [30] although archaeal EF-2 gene has been widely used as a phylogenetic marker for archaea. We confirmed that LAMP primer sets reliably amplified archaeal EF-2 from selected genera of hydrogenotrophic methanogens. Although some of our primers aligned to eukaryotic EF-2 genes, which could lead to false-positive results for environmental samples containing highly abundant archaeal and eukaryotic species, no amplification was observed using environmental DNA extracted from a wheat root rhizobiome, suggesting that our designed primer sets are specific for the targeted archaeal EF2 genes.

In testing, our LAMP assay successfully detected *Methanoculleus* species in samples from process-scale wastewater biosolids digesters (Figure 5a, lanes 1, 2) consistent with metagenomic sequencing results of the same samples (Alessi et al., in preparation) and as expected from studies using different detection methods [31]. *Methanoculleus* species are common to bioreactors operating at high ammonia concentrations [32], typically dominating anaerobic digestion communities in reactors processing manures [33]. Consistent with other literature [8,34] we also detected *Methanoculleus* in lab-scale reactors supplemented with hydrogen during in situ CO_2_ biomethanisation (Figure 5a, lanes 4, 5) but not in thermophilic food waste fed systems (Figure 5, lanes 6, 7), indicating our primers are specific for mesophilic *Methanoculleus* species. Additionally, our LAMP assay results are consistent with 16S rRNA amplicon profiles obtained for these samples in our previous work. We showed that CO_2_ reductive biomethanisation systems were dominated by *Methanoculleus* genus hydrogenotrophs and that the abundance of these species increased with a rise in H_2_ addition to these systems [5]. 

We detected *Methanothermobacter* in lab-scale digesters treating food waste at 55 °C using LAMP. *Methanothermobacter* species are generally found in thermophilic anaerobic digestate with growth temperatures ranging from 40 to 70 °C [35,36].

Our results show our assays are able to differentiate other hydrogenotrophic genera such as *Methanococcus* [37] and *Methanobrevibacter.* Although our positive control indicated assay specificity, we were unable to detect *Methanococcus* in the available salt marsh samples, possibly due to low microbial abundance resulting in DNA concentrations below the sensitivity of our LAMP assay. Consistent with 16S rRNA amplicon sequencing [5], *Methanococcus* was not detected by LAMP in mesophilic lab scale digesters (Figure 5c, lanes 4, 5). Using DNA extracted from stool samples from human volunteers, we confirmed the presence of *Methanobrevibacter* by LAMP. *Methanobrevibacter* species are typically associated with the gastrointestinal tracts of herbivorous ruminants, termites, and humans [38,39,40]. Thus, our LAMP assays could potentially be used in a clinical setting as an alternative to a methane breath test.

The LAMP assays described here offer a simple, fast, and affordable method for the specific detection of methanogens in anaerobic digestion and other samples. Given the different thresholds of sensitivity shown in Figure 6, our results indicate that with further work this approach could be developed into a rapid quantitative measure. For example, using quantified reference standards and serial dilutions, genera could be quantified to an order of magnitude in samples for less than 1 USD in an hour or less. Previous LAMP-based studies have focused on the detection of single species, whereas we have designed LAMP assays spanning three different orders (*Methanobacteriales*, *Methanococcales,* and *Methanomicrobiales*) and four different hydrogenotrophic genera that should provide additional rapid, low-cost insight into the functioning of anaerobic digestion and related systems. Such insights in turn will provide a basis for guidance of process operations and help to support the development of new areas of application from reductive CO_2_ biomethanisation to novel methane-based biorefineries.

## Figures and Tables

**Figure 1 microorganisms-08-00740-f001:**
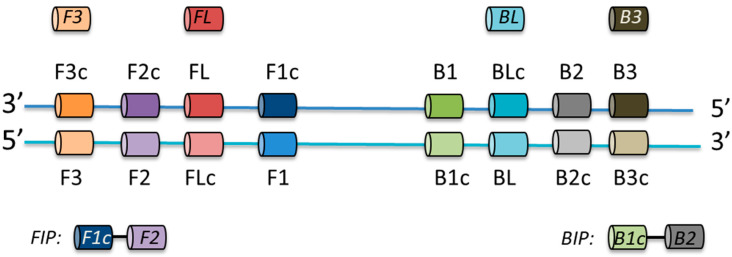
Cartoon outlining the relative positioning and relationships between the primers required for a LAMP assay.

**Figure 2 microorganisms-08-00740-f002:**
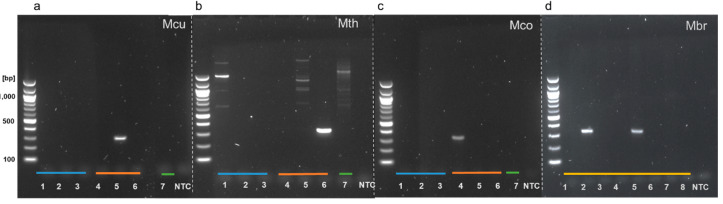
Agarose gel electrophoresis analysis of PCR products of reactions performed with F3 and B3 outer primers and selected bacterial, archaeal, and fungal DNA as templates. All the components in the PCR reactions were the same except for F3/B3 primers that were (**a**) *Methanoculleus* (Mcu), (**b**) *Methanothermobacter* (Mth), (**c**) *Methanococcus* (Mco), and (**d**) *Methanobrevibacter* (Mbr). Bacterial DNA templates (blue line) from *Escherichia coli* (lane 1), *Staphylococcus epidermidis* (lane 2), and *Bacillus* sp. 3PL (lane 3). Archaeal DNA templates (orange line) from *Methanococcus maripaludis* (lane 4), *Methanoculleus marisnigri* (lane 5), and *Methanothermobacter thermoautotrophicus* (lane 6), eukaryotic DNA was extracted from wheat root rhizobiome (green line, lane 7). For (d), eDNA from stool samples was used as a template (yellow line, lanes 1–8). A non-template control (NTC) was included in all assays.

**Figure 3 microorganisms-08-00740-f003:**
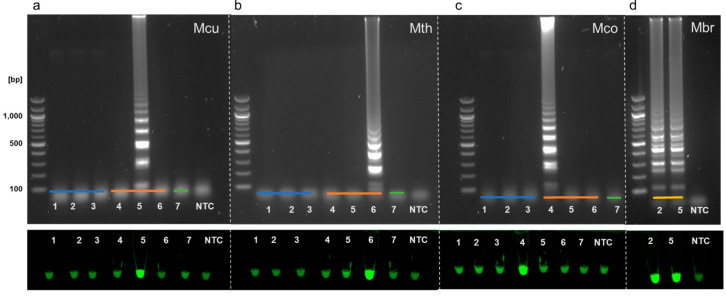
LAMP assay outputs comparing agarose gel electrophoresis (top of panels) and Sybr Green I end-point detection (bottom of panels). Specific F3/B3 primers and genomic DNA from (**a**) *Methanoculleus*, (**b**) *Methanothermobacter*, (**c**) *Methanococcus,* and (**d**) *Methanobrevibacter* were used in assays performed with either Bst 3.0 (**a**,**c**,**d**) or Bst 2.0 (**b**). Bacterial DNA (blue line) originated from *Escherichia coli* (lane 1), *Staphylococcus epidermidis* (lane 2), and *Bacillus* sp. 3PL (lane 3). Archaeal DNA (orange line) originated from *Methanococcus maripaludis* (lane 4), *Methanoculleus marisnigri* (lane 5), and *Methanothermobacter thermoautotrophicus* (lane 6), eukaryotic DNA was extracted from wheat root rhizobiome (green line, lane 7). For (**d**), eDNA extracted from two *Methanobrevibacter* positive (see Figure 2) stool samples was used as a template (yellow line). A non-template control (NTC) was included in all assays.

**Figure 4 microorganisms-08-00740-f004:**
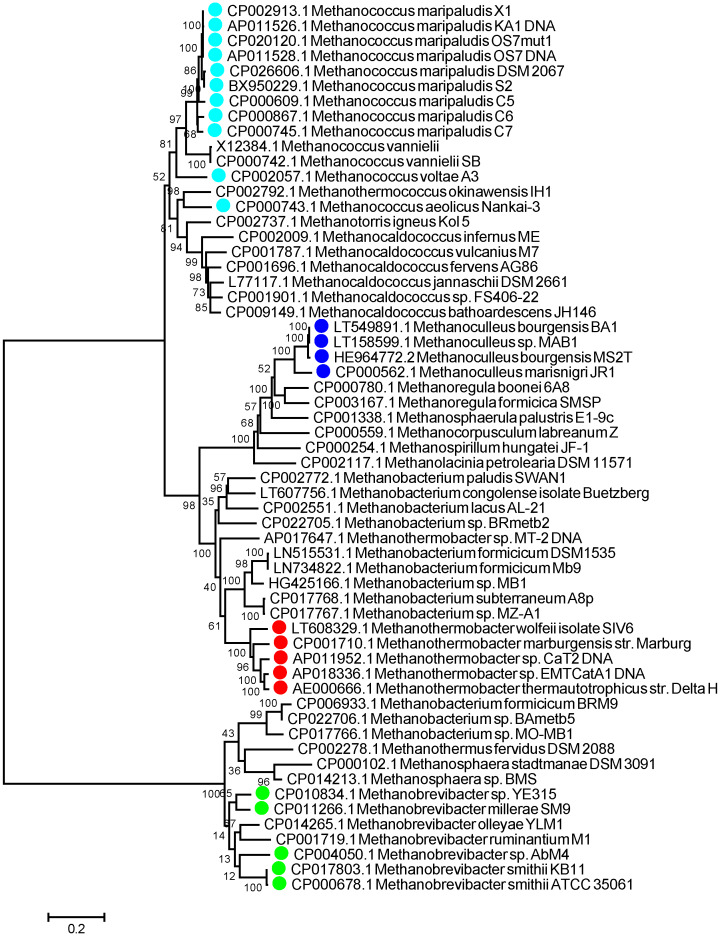
Specificity of LAMP primer sets for targeted genera within the methanogenic archaea. Evolutionary relationships of elongation factor 2 (aEF-2) genes within hydrogenotrophic methanogens were conducted in MEGA7 and inferred using neighbour-joining. Evolutionary distances were computed using the Jukes–Cantor method as number of base substitutions per site. The analysis used 59 nucleotide sequences. All positions containing gaps and missing data were eliminated. There were a total of 1971 positions in the final dataset. An optimal tree with sum of branch length = 6.51376856 is shown. The percentage of replicate trees in which the associated taxa clustered together in the bootstrap test (based on 500 replicates) are annotated branches. The tree is drawn to scale, with branch lengths in the same units as the evolutionary distances used to infer the phylogenetic tree. LAMP primer sets predicted to target specific species are shown as coloured circles (cyan: *Methanococcus*; blue: *Methanoculleus*, red: *Methanothermobacter*; green: *Methanobrevibacter*).

**Figure 5 microorganisms-08-00740-f005:**
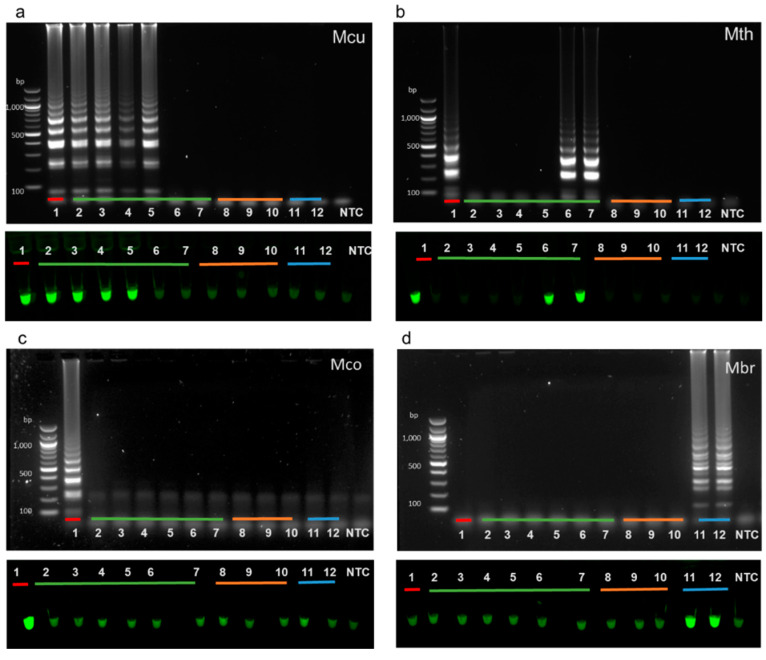
LAMP assay outputs comparing agarose gel electrophoresis (top of panels) and Sybr Green I end-point detection (bottom of panels). Assays used primers targeted to species with the genera *Methanoculleus* (**a**), *Methanothermobacter* (**b**), *Methanococcus* (**c**), and *Methanobrevibacter* (**d**). Assays were performed with Bst 3.0 (**a**,**c**,**d**) or Warm Start Bst 2.0 chemistry (**b**). Genomic DNA (20 ng) from *Methanoculleus* (**a**)*, Methanothermobacter* (**b**), or *Methanococcus* (**c**) was used as a positive control (lane 1, red line). LAMP assays were performed with metagenomic DNA from process scale mesophilic digesters (lanes 2, 3), lab-scale mesophilic digesters (lanes 4, 5), lab-scale thermophilic anaerobic digesters (lanes 6, 7; green line), salt marsh sediment (lanes 8, 9; orange line), garden soil (lane 10; orange line), or human stool samples (lanes 11, 12; blue line). A non-template control (NTC) was included in all assays.

**Figure 6 microorganisms-08-00740-f006:**
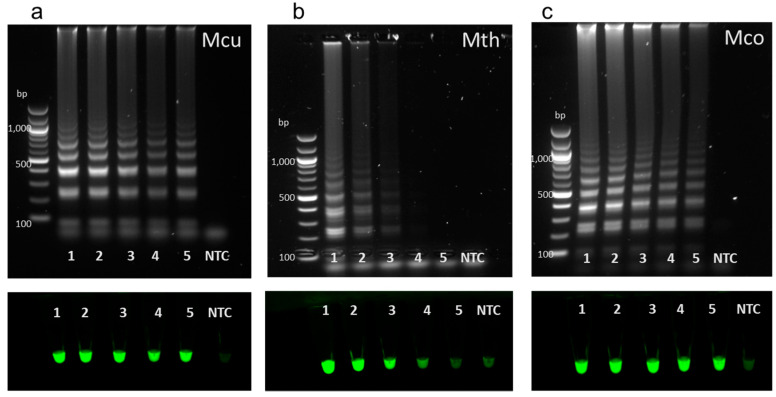
LAMP assay outputs comparing agarose gel electrophoresis (top panels) and Sybr Green I end-point detection (bottom panels). Genomic DNA from (**a**) *Methanoculleus marisnigri* JG1 (Mcu), (**b**) *Methanothermobacter thermautotrophicus* (Mth), and (**c**) *Methanococcus maripaludis* S1 (Mco) were added in decreasing quantities (lane 1: 10 ng, lane 2: 1 ng, lane 3: 100 pg, lane 4: 10 pg, lane 5: 1 pg) to genus-specific LAMP assays and compared to non-template controls (lane NTC) to determine assay sensitivity.

**Table 1 microorganisms-08-00740-t001:** Loop-mediated isothermal amplification (LAMP) primer sequences targeting various hydrogenotrophic methanogen species.

Target Genus	Oligo Name *	Oligo Sequence (5′ to 3′) ^#^	Tm [°C]	Length [bp]
***Methanoculleus*** **(Mcu) ^a^**	F3_aEF2_Mcu	TAYCTBATCAACATGATYGAT	48.0	21
	FLc_aEF2_Mcu	ACCGCRTCCACSACRAC	58.4	17
	FIP_aEF2_Mcu	*GTCTCCGTCTGKGGCATGGT* **TTTT** GACGTGACYCGYGCCATG	70.9	42
	B3_aEF2_Mcu	TYYTCGTTCATDCCCTTGAT	52.3	20
	BL_aEF2_Mcu	ACGAGCAGGAGATGCAGATC	56.8	20
	BIP_aEF2_Mcu	*ACCGGCTGRTCAACGAG* **TTTT** TTGACCTTRTCGATCAC	65.8	38
***Methanothermobacter*** **(Mth) ^b^**	F3_aEF2_Mth	ATTAAGGAGCTCATGTACCA	50.6	20
	FLc_aEF2_Mth	CAAGGAARCGCTGATCCC	54.9	18
	FIP_aEF2_Mth	*CCTTGCCTGTTCCTGTTC* **TTTT** GACAACCTCCTGGCTGGTGC	69.7	42
	B3_aEF2_Mth	GCATTATRCCCTCAACKGC	54.0	19
	BL_aEF2_Mth	TCAATGGTCCACTCCTA	49.1	17
	BIP_aEF2_Mth	*ACCATTGACGCTGCRAACGT* **TTTT** CCTCATKGCCCTTGTAACGT	69.1	44
***Methanococcus*** **(Mco) ^c^**	F3_aEF2_Mco	ATGGGAAGAAGAGCAAAAATG	51.3	21
	FLc_aEF2_Mco	ATCATWCCWGCWCCTGC	52.0	17
	FIP_aEF2_Mco	*CAAGYTGGTCTCCWGC* **TTTT** CATYGACCACGGTAAAAC	65.1	38
	B3_aEF2_Mco	CTCATTGCTCTTGTAACGTC	51.1	20
	BL_aEF2_Mco	CTGCAAACGTKTCAATGGT	52.7	19
	BIP_aEF2_Mco	*GAAGAAGCWGCAAGAGGTAT* **TTTT** TCAACGTGACCWGGGGTRTC	66.4	44
***Methanobrevibacter*** **(Mbr) ^d^**	F3_aEF2_Mbr	GAAACTGTAYTYAGACAA	43.5	18
	FLc_aEF2_Mbr	TCAGGAGCCATGTTYTTGATTA	53.1	22
	FIP_aEF2_Mbr	*GCTGAACCRAAWGCTACACT* **TTTT** AATCAACGARTTAAAATTA	60.6	43
	B3_aEF2_Mbr	AAGTGTTCTACTACCATAC	45.6	19
	BL_aEF2_Mbr	ATYATTGATTAYTGTAATG	39.4	19
	BIP_aEF2_Mbr	*TGGGCTATYAAYGTTCC* **TTTT** GGTACTTTTTKAGCTAATTC	61.7	41

**^a^** CP000562.1: *Methanoculleus marisnigri* JR1, NC_018227.1 (HE964772.2): *Methanoculleus bourgensis* MS2T, LT158599.1: *Methanoculleus* sp. MAB1, LT549891.1: *Methanoculleus bourgensis* isolate BA1. **^b^** CP001710.1 (NC_014408.1): *Methanothermobacter marburgensis* str. Marburg, AP011952.1: *Methanothermobacter* sp. CaT2, AE000666.1 (NC_000916.1): *Methanothermobacter thermautotrophicus* str. Delta H, LT608329.1: *Methanothermobacter wolfeii* isolate SIV6, **^c^** CP026606.1: *Methanococcus maripaludis* strain DSM 2067, CP002913.1: *Methanococcus maripaludis* X1, CP000609.1: *Methanococcus maripaludis* C5, CP000745.1: *Methanococcus maripaludis* C7, CP000867.1: *Methanococcus maripaludis* C6, CP002057.1: *Methanococcus voltae* A3, **^d^** CP000678.1: *Methanobrevibacter smithii* ATCC 35061, CP004050.1: *Methanobrevibacter* sp. AbM4, CP010834.1: *Methanobrevibacter* sp. YE315. * F3: forward outer primer, FL: forward loop primer, FIP: forward inner primer that consists of the F2 region (at the 3′ end) complementary to the F2c region, and the same sequence as the F1c region at the 5′ end, B3: backward outer primer, BL: backward loop primer, BIP: backward inner primer consists of the B2 region (at the 3′ end) complementary to the B2c region, and the same sequence as the B1c region at the 5′ end. #: F(B)1c primer is in italics, linker is in bold, and F(B)2 primer is underlined in the F(B)IP primers.

**Table 2 microorganisms-08-00740-t002:** Analysis of potential gene targets for the LAMP assay for detection of hydrogenotrophic methanogens in environmental samples.

	ID (%)	# Sequences (Uniprot)	# Identical Positions	# Similar Positions
***fusA* (aEF-2) (K03234)**				
***Methanomicrobiales***	61	9	449	157
***Methanobacteriales***	47	24	356	192
***Methanococcales***	61	17	447	172
***mcrA* (K00399)**				
***Methanomicrobiales***	57	15	325	132
***Methanobacteriales***	45	41	250	157
***Methanococcales***	66	22	366	117
***mer* (K00320)**				
***Methanomicrobiales***	63	9	212	70
***Methanobacteriales***	34	24	110	105
***Methanococcales***	58	17	192	75
***ehaA* (K14097)**				
***Methanomicrobiales***	36	6	63	38
***Methanobacteriales***	23	21	48	51
***Methanococcales***	36	17	62	40
***rsp28e* (K02979)**				
***Methanomicrobiales***	75	9	52	10
***Methanobacteriales***	63	24	46	16
***Methanococcales***	57	17	44	22
***rpl44e* (K02929)**				
***Methanomicrobiales***	61	9	56	18
***Methanobacteriales***	53	24	50	24
***Methanococcales***	51	17	49	15

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
