# Peer review of "A Rapid, Sensitive, Low-Cost Assay for Detecting Hydrogenotrophic Methanogens in Anaerobic Digesters Using Loop-Mediated Isothermal Amplification"

_microorganisms, 2020, doi:10.3390/microorganisms8050740_

Round 1

Reviewer 1 Report

This paper deals with a very interesting issue. It is basically well written. However to a reader non-expert in molecular biology (such as most of the professionals involved in AD) the simplicity and low cost of this technique are far from being evident. In addition you have to stress that this is only a first step towards a quantitative test. Also if not completely useless a test presence/absence would be infact of limited use.

Line 44: please avoid the word “destruction”, “degradation” is more correct in this context

Lines 49-63: simplified equations for methane formation using different substrates could help the reader

Line 65: you have to specify the measurements you are referring to

Author Response

  • This paper deals with a very interesting issue. It is basically well written. However to a reader non-expert in molecular biology (such as most of the professionals involved in AD) the simplicity and low cost of this technique are far from being evident. In addition you have to stress that this is only a first step towards a quantitative test. Also if not completely useless a test presence/absence would be infact of limited use.

Response: We have modified the text in a number of places to emphasise that LAMP assays are rapid (lines 28, 425-7) low cost (lines 28, 100-1, 425-7) and require little equipment (line 100) to support our previous statements (lines 93-99, 370-2). We have added information concerning how these assays could be made more quantitative (lines 31, 425-7).

  • Line 44: please avoid the word “destruction”, “degradation” is more correct in this context

Response: The text was modified as requested by changing “destruction” to “degradation”.

  • Lines 49-63: simplified equations for methane formation using different substrates could help the reader

Response: This paragraph was modified as suggested by the reviewer and now includes additional text concerning different methanogenesis pathways and their relevant equations

  • Line 65: you have to specify the measurements you are referring to

Response: We have expanded the relevant sentence to include the following: “measurements indicating the presence / absence of particular microbial community members, or that quantify these”

Reviewer 2 Report

The manuscript gives new information on detection of four hydrogenotrophic methanogens using the LAMP assays. However, this paper has some problems. I think that it requires major revision for publication.

  1. Abstract

In abstract, summary of this article is not described. Authors should rewrite the abstract with important results.

  1. Introduction

  In introduction, authors does not make mention of four methanogens, Methanoculleus spp, Methanothermobact spp, Methanobrebioacter spp. and Methonococcus spp. They should add the function of four methanogens in anaerobic digestion and importance of detecting four methanogens.

  1. Authors should describe four methanogens in the same order.

  1. 2.5 LAMP assay

  Visual detection LAMP assays were reported by many researchers. They performed the LAMP assay with various dyes. Why didn’t authors use these dyes?

  1. 3.2 Genus-specific PCR products are generated with LAMP outer primers

  I cannot understand meaning of this section.

  1. L294-

  Authors didn’t use pure sample of Methanobrevibacter only environmental samples. They should show the proof that this LAMP assays detect this genera.

  1. Fig4

  Authors show LAMP detection results using various samples. They show that LAMP assays detect these genera correctly by adding other detection test data.

  1. L79

  ‘six to eight primers’ to ‘four to six primers (from six to eight regions)’

  1. L80

  In many reports, the LAMP reactions were performed for one hour. So ‘ less than 30 minutes’ should be changed to ‘less than one hour’.

  1. Fig4 L304

   Correct ‘Methanobrevibacter’ and ‘Methanococcus’.

Author Response

The manuscript gives new information on detection of four hydrogenotrophic methanogens using the LAMP assays. However, this paper has some problems. I think that it requires major revision for publication.

  • Abstract: In abstract, summary of this article is not described. Authors should rewrite the abstract with important results.

Response: we have revised the abstract to details of the LAMP assays we have developed including target genera, the gene used for differentiation and levels of sensitivity.  We have identified cost and speed as two advantages of this measurement.

  • Introduction: In introduction, authors does not make mention of four methanogens, Methanoculleus spp, Methanothermobact spp, Methanobrebioacter spp. and Methonococcus spp. They should add the function of four methanogens in anaerobic digestion and importance of detecting four methanogens.

Response: Additional information on the function of these methanogens in AD was added to the introduction with references to the relevant studies. The importance of their detection was already included in the text in lines 111-113.

  • Authors should describe four methanogens in the same order.

Response: We agree that this is a logical format and have corrected the manuscript where appropriate to apply this rule throughout.

  • 5 LAMP assay: Visual detection LAMP assays were reported by many researchers. They performed the LAMP assay with various dyes. Why didn’t authors use these dyes?

Response: As reported in numerous other studies and described in section 2.5, we have used the SYBR Green I dye.  This provides a colour change that is visible by eye, although enhanced under fluorescence.  These results are shown in Figures 2, 4 & 5.

  • 2 Genus-specific PCR products are generated with LAMP outer primers: I cannot understand meaning of this section.

Response: We apologise that this section is confusing to the reviewer. We intended to demonstrate that the outer primers were unique and specific, so that we do not produce off-target products. We sequenced the resulting PCR product which confirmed that only the aEF2 genes from targeted methanogens was amplified. We have amended text of this section describing why this work was carried out and included the sequencing results in Table S1.

  • L294 - Authors didn’t use pure sample of Methanobrevibacteronly environmental samples. They should show the proof that this LAMP assays detect this genera.

Response: As originally noted in our manuscript, a pure culture of Methanobrevibacter was not available to us during this study. This situation in likely to occur in AD, where a number of methanogenic species are not available as pure cultures, but it would still be useful to be able to detect them in a rapid and cost-effective manner. To address this point, we used DNA extracted from stool samples from eight different individuals. It is estimated that at least 1 in 3 of the population are methanogenic and in these cases Methanobrevibacter has been reported as part of the human gut microbiome. We performed the initial test with outer primers for this genus and showed successful amplification of the aEF2 gene in two samples using our F3/B3 MBr primers (consistent with estimated population frequencies). To confirm that the primers were specific towards this genus and aEF2 gene, we sequenced the amplified target which was consistent with those found in Methanobrevibacter species.

We have included this data in Figures 1, 2 and Table S1 and modified the text accordingly.

  • Fig4 - Authors show LAMP detection results using various samples. They show that LAMP assays detect these genera correctly by adding other detection test data.

Response: Thank you for this comment.

  • L79 - ‘six to eight primers’ to ‘four to six primers (from six to eight regions)’

Response: We have corrected this sentence. For additional clarity, we have added a reference to the figure in Table 1. We have also updated Table 1 to include standard nomenclature for the primers’.

  • L80 - In many reports, the LAMP reactions were performed for one hour. So ‘ less than 30 minutes’ should be changed to ‘less than one hour’.

Response: We have modified this sentence in accordance with the reviewer’s request.

  • Fig4 L304 - Correct ‘Methanobrevibacter’ and ‘Methanococcus’.

Response: Thanks for spotting this mistake. The legend for Figure 4 has been corrected to reflect the panels correctly.

Round 2

Reviewer 1 Report

The paper can be accepted

Author Response

Thank you for your time and support.

Reviewer 2 Report

  1. In each table and figure, authors should describe four methanogens in the same order.

Table1, Fig.4 : Mcu→ Mth→ Mco→ Mbr

Fig.1, Fig.2: Mco→ Mcu→ Mth→ MBr (Mbr)

  1. 2.5 LAMP assay

Visual detection LAMP assays were reported by many researchers. They performed the LAMP assay using the reaction solution including various dyes (e.g. HNB, calcein). Why didn’t authors use these dyes?

  1. Figure 2(d)

Add the LAMP results of other eDNA samples to prove the specificity of the LAMP reaction.

  1. Fig4

Add the PCR results of same DNA samples to prove the specificity of the LAMP reaction.

Author Response

We thank this reviewer for their time and detailed comments which we address below:

In each table and figure, authors should describe four methanogens in the same order.

Table1, Fig.4 : Mcu→ Mth→ Mco→ Mbr

Fig.1, Fig.2: Mco→ Mcu→ Mth→ MBr (Mbr)

Response: We apologise for not completing this request satisfactorily in our original revisions. As requested, the displayed data is now all in the same order. Specifically, Figures 1 and 2 have been modified to match the order Mcu-Mth-Mco-Mbr of Table 1 and Figure 4. Figure 2 – the “MBr” label was modified to “Mbr”.

2.5 LAMP assay

Visual detection LAMP assays were reported by many researchers. They performed the LAMP assay using the reaction solution including various dyes (e.g. HNB, calcein). Why didn’t authors use these dyes?

Response: It is not clear why this reviewer keeps asking this question as it has little-to-no bearing on the outcome of the assays or the substance of the results or manuscript.

There are two main approaches to detecting LAMP assay end products: indirect and direct detection of reaction (by-)products:

  1. Indirect detection of reaction by-products is based on the release of pyrophosphate during the incorporation of deoxyribosenucleoside triphosphates (dNTPs) into nascent DNA. This can be detected either by visual observation of turbidity or by the addition of dyes such as HNB or calcein to detect the LAMP by-products.
  2. Direct detection uses intercalating dyes such as SYBR Green I and EvaGreen which strongly bind to dsDNA and are therefore ideal candidates for the detection of nascent DNA in LAMP reactions.

The choice of dye is largely irrelevant to the outcome of the assay and is a matter of individual choice. We have opted for a direct detection method for our LAMP assay products because of its simplicity, sensitivity and availability of reagents. There is the potential for indirect detection dyes to respond to contaminants in the reactions resulting in false positives. Direct detection occurs only on amplification of DNA, but would respond equally to off-target products.  We have demonstrated that our primers produce on-target products (Figures 1 & 2). As highlighted by the reviewer, indirect detection could also be applied to these reactions. We do not feel it is appropriate to provide a detailed critique of LAMP detection methods in this manuscript as there are many reviews discussing these details. It is not clear what action needs to be taken to address this point in the manuscript.

Figure 2(d)

Add the LAMP results of other eDNA samples to prove the specificity of the LAMP reaction.

Response: This is a new request following the original review. LAMP results for a range of other eDNA samples can be found in Figure 4d where we tested samples from mesophilic and thermophilic anaerobic digesters, salt marsh sediment, garden soil and human stools. We have already explained that we did not have access to pure DNA for this genus and understand that this is a less-than-perfect proof of the specificity of the LAMP reaction for Methanobrevibacter species, which is why we highlighted this point originally. At this reviewer’s request, we had already expanded on this point in the revised manuscript (lines 264-267).

Fig4

Add the PCR results of same DNA samples to prove the specificity of the LAMP reaction.

Response: This is a new request following the original review. It is not clear what the reviewer requires and hence is difficult to address. Is the reviewer requesting that we repeat the experiment carried out in Figure 1 with mixed samples? How will this help address specificity without sequencing the products, which are likely to be a mixture of different species and therefore require additional separation, probably by cloning? In Figures 1 and 2 we have demonstrated that the primers we have designed show good specificity for the genera we are trying to target. The ladder patterns observed in Figure 4 are highly comparable to those in Figure 2, suggesting that the LAMP products are the same as those observed for purified DNA (where it was available).